# Infrared Thermography as a Method of Verification in Raynaud’s Phenomenon

**DOI:** 10.3390/diagnostics11060981

**Published:** 2021-05-28

**Authors:** Lotte Lindberg, Bent Kristensen, Ebbe Eldrup, Jane Frølund Thomsen, Lars Thorbjørn Jensen

**Affiliations:** 1Department of Nuclear Medicine, Copenhagen University Hospital, Herlev Hospital, Borgmester Ib Juuls Vej 1, 2730 Herlev, Denmark; bent.kristensen.01@regionh.dk (B.K.); lars.thorbjoern.jensen@regionh.dk (L.T.J.); 2Department of Endocrinology, Copenhagen University Hospital, Herlev Hospital, Borgmester Ib Juuls Vej 1, 2730 Herlev, Denmark; ebbe.eldrup@regionh.dk; 3Department of Occupational and Environmental Medicine, Copenhagen University Hospital, Bispebjerg and Frederiksberg Hospital, Bispebjerg Bakke 23, 2400 Copenhagen, Denmark; jane.froelund.thomsen@regionh.dk

**Keywords:** cold challenge, diagnostic method, infrared thermographic imaging, secondary Raynaud’s phenomenon, vibration white finger

## Abstract

Raynaud’s phenomenon (RP) is characterized by the episodic whitening of the fingers upon exposure to cold. A recently described thermographic algorithm was proposed as a diagnostic replacement of the currently applied finger systolic pressure (FSP) test. The aim of the study was to evaluate the performance of the thermographic algorithm when applied in patients suspected of having RP. Forty-three patients were examined using thermographic imaging after local cooling of the hands in water of 10 °C for 1 min. The thermographic algorithm was applied to predict the probability of RP. The performance of the algorithm was evaluated with different cut-off levels. A new algorithm was proposed based on patients from the target population. The performance of the tested algorithm was noninferior to the FSP test, when a cut-off level of 0.05 was applied, yielding a sensitivity and specificity of 69% and 58%, respectively. The accuracy was 66%. The FSP test had a sensitivity and specificity of 77% and 37%, respectively, and the accuracy was 59%. The thermographic method proved useful for detecting RP and was able to replace the FSP test as a diagnostic test. The alternative algorithm revealed that other thermographic variables were more predictive of the target population, but this should be verified in future patients.

## 1. Introduction

Raynaud’s phenomenon (RP) is characterized by the episodic whitening of the fingers upon exposure to cold. Although the cause cannot be identified in most patients, RP can be induced by certain drugs, an underlying rheumatic disease, or repeated exposure to hand–arm vibrations [1]. RP is typically a clinical diagnosis based on medical history, a physical examination, and blood tests. However, RP symptoms are not always characteristic. In vibration white finger (VWF), which is caused by occupational exposure to vibration, the diagnosis must be verified in order for the condition to be accepted as an occupational injury. Furthermore, a correct diagnosis is important for the patient, as continued exposure may aggravate the disease [2], possibly resulting in recommendations to change professions. The diagnosis is verified if an episode is witnessed by a physician or the patient is able to capture a photo of an episode, either of which has proven easy to obtain. As a result, an objective examination that includes a cold provocation test is often required to substantiate the diagnosis.

The method of systolic blood pressure with cold provocation is currently applied in our department and has been used for decades [3]. Under this method, the decrease in finger systolic blood pressure is measured after 10 min of total body cold exposure and local finger cooling. This method is cumbersome, time-consuming, and unpleasant for the patient. Moreover, despite obvious anamnestic signs of RP, the method sometimes neglects to detect the disease. Additionally, the method requires outdated technical equipment no longer supported by the manufacturer, which makes it necessary to develop a new procedure for detecting RP.

Infrared thermographic imaging has been the subject of numerous investigations that include various patient groups, purposes, and outcomes. The performances of the different methods vary widely and seem to depend on the prevalence of RP in the study population, the severity of the symptoms [4,5], and the possibility of closely controlling the environmental conditions during the examination [4,6,7]. We recently investigated infrared thermographic imaging as a method for distinguishing patients with primary Raynaud’s phenomenon (pRP) from healthy individuals [8]. The study generated a prediction model that was able to predict the presence or absence of RP with a sensitivity of 82% and a negative predictive value (NPV) of 93% and was not inferior to the previous studies on thermographic imaging [5,9,10,11]. Thus, infrared thermographic imaging showed great promise as a verification method in patients with RP, where confirmation of the diagnosis is needed.

The aim of the present study was to test the newly described thermographic method in patients with RP, primarily associated with occupational exposure to vibration and whose diagnosis was not yet verified, and compare the performance of the method with that of the currently applied method. The hypothesis was that the new method would not be inferior to the finger systolic pressure (FSP) test in performance and able to replace this method as a diagnostic test. In addition, we wanted to explore whether the thermographic parameters characteristic for pRP and VWF, respectively, were similar.

## 2. Materials and Methods

### 2.1. Participants

Forty-three patients with suspected RP were included in the study. Thirty-five patients were referred from two Departments of Occupational and Environmental Medicine in Eastern Denmark due to occupational exposure to vibration and suspected VWF. The remaining eight patients were referred from other medical specialties, including general practitioners, and had not been exposed to vibration. Thus, the origin of RP was unknown for this subgroup. None of the patients had diabetes mellitus, were previously treated with chemotherapy, or were diagnosed with vitamin B_12_ deficiency or renal impairment. Twenty-two patients had already been examined using the existing diagnostic method (FSP test) from January 2014 to October 2016. The remaining 21 patients were recruited immediately after an examination with the FSP test (from January 2017 to January 2019). The two groups of patients were named the retrospective and the prospective groups, respectively, and were included consecutively.

For the purpose of evaluating the performance of the algorithm in the patients suspected of RP, the clinical diagnosis was used as the reference standard. Thus, the clinical diagnosis was made by two senior consultants upon review of the medical records of 40 patients (three were missing). The two readers independently assigned a diagnosis (positive or negative of RP) to each patient blinded to the results of the FSP and thermographic tests. Eight diagnostic disagreements between the readers were discussed, and joint diagnoses were made. The decision, which was based solely on the symptoms, physical examination, and blood test results described in the medical records, served as the true diagnosis of RP. The medical records of three patients were missing.

The control group was identical to the control group described in reference [8]. The healthy participants were recruited through advertisements in Herlev and Gentofte Hospital and in local newspapers. All 58 participants included in the control group had no symptoms of RP or history of occupational exposure to vibration. Moreover, they had no medical record of chronic illnesses or health issues affecting the blood vessels, especially heart or lung diseases, or connective tissue diseases. They did not have vitamin B_12_ deficiency or alcohol abuse, they were alleged nonsmokers (one participant changed smoking status after inclusion), and they did not take prescription drugs. Thus, 101 subjects were examined using the thermographic method.

Blood tests were taken in order to exclude differential diagnoses to RP: folic acid, vitamin B_12_, hemoglobin, albumin, ionized calcium, potassium, magnesium, sodium, glucose, HbA1c, TSH, CRP, and vitamin D. The examinations were performed at the Department of Nuclear Medicine, Copenhagen University Hospital, Herlev Hospital, from January 2017 to February 2019.

### 2.2. Thermographic Imaging Procedure and Data Analysis

The thermographic procedure, preparations, and subsequent analysis were conducted as previously described [8]. In short, the participants’ hands were immersed in water of 10 °C for 60 s, and the rewarming period of the fingers was recorded using an infrared thermographic camera (FLIR SC600, FLIR^®^ Systems, Wilsonville, OR, USA). The thermographic recordings resulted in rewarming curves, which were analyzed to obtain the seven temperature variables previously described [8]: baseline finger temperature (t_base_), finger temperature immediately after cooling (t_0_), curve type, finger temperature halfway through rewarming (t_50_), end temperature (t_end_), time to t_end_, and the percentage of temperature recovery at t_end_ (R%) (R% = temperature increase/initial temperature decrease) × 100% [12]. The categorical variable *curve type* described the course of the temperature curve as either S-shaped or horizontal. See Figure 1 for an example of each curve type with the associated thermographic images at the essential time points.

### 2.3. Finger Systolic Pressure (FSP) Test

The currently applied FSP procedure uses body cooling, as well as local cooling, to 30 °C and 10 °C to provoke an episode of RP. At 10 °C, the discriminating threshold for the diagnosis of RP is defined as an FSP of 20% or less of the FSP at 30 °C, while an abnormal cold response is defined as an FSP of 20–63% of the FSP at 30 °C. An FSP above 63% of the FSP at 30 °C is defined as a normal response.

### 2.4. Statistics

Descriptive statistics were presented as the mean and standard deviation or median and interquartile range (IQR) where appropriate. Counts were presented as numbers and proportions. For normally distributed variables, group comparisons were done using the Student’s *t*-test. For non-normally distributed variables, comparisons were done using the Wilcoxon rank-sum test. Due to the small cell counts, categorical variables were compared groupwise by applying Fisher’s exact test.

The coefficients of the predictor variables (*time to t_end_* and *t_base_*) (Table 1) and the respective values of *time to t_end_* and *t_base_* generated from the thermographic examination of the patients suspected of RP were used to calculate the patients’ respective predicted probabilities of having RP. Different cut-off values of the predicted probability were assessed, including the cut-off value originally chosen. The performances of the thermographic algorithm and the FSP test were compared using superiority and noninferiority analyses, where a margin of Δ = –15% was chosen.

After the evaluation of the previously developed thermographic method, a new algorithm was generated to investigate if pRP and VWF share the same thermographic characteristics. The fitting of a new logistic prediction model based on the thermographic parameters of the patients positive for RP in the present cohort and the healthy controls and evaluation of the model were conducted as described previously [8]. All the statistical calculations were done using the statistical software R, version 4.0.2, June 2020), (R Core Team (2020). R: A language and environment for statistical computing. R Foundation for Statistical Computing, Vienna, Austria. URL https://www.R-project.org/), and the following packages: *car*, *ClusterRankTest*, *confinterpret*, *growthrates*, *Hmisc*, *pROC*, *PropCIs*, *rms*, and *tableone*.

## 3. Results

### 3.1. Clinical Characteristics

Two patients and one healthy individual were excluded from the analyses, as their respective examinations ended prematurely (due to back pain and technical problems). Accordingly, the results from 98 subjects were included in the analyses. The participants consisted of 27 (27.6%) women and 71 (72.4%) men. Their clinical characteristics are presented in Table 2.

As defined in the exclusion criteria, none of the healthy participants were repeatedly exposed to vibration, while 34 (83%) of the patients were exposed to vibration in relation to their occupation. Two (5%) patients were treated for RP with drugs. In both cases, the treatment was paused 24 h prior to examination.

The patients in the retrospective group had symptoms of RP significantly longer than the patients in the prospective group (median (IQR): 10 years (8; 15) and 5 years (3; 8), respectively). Furthermore, the retrospective group had a higher alcohol intake, as well as a larger proportion of patients currently smoking, and, thereby, a higher consumption of tobacco than the prospective group. Although the potassium level was within the normal limits, the level was slightly higher in the retrospective group than in the group of healthy individuals.

The blood tests showed no differences between the patients and the healthy individuals in terms of folic acid, albumin, ionized calcium, magnesium, sodium, glucose, HbA1c, TSH, CRP, and Vitamin D. The hemoglobin was significantly lower in the control group, which was probably due to the larger proportion of female participants in this group (42%) compared with the patient group (7%). Vitamin B_12_ was significantly lower and potassium significantly higher in the patient group than in the control group. The mean plasma level of vitamin B_12_ was above the lower reference limit (>200 pmol/L). However, for five patients and one healthy individual, their levels of vitamin B_12_ were lower than the commonly used reference limit.

Based on the clinical decisions, a total of 26 of the 38 patients (68%) were diagnosed with RP, while the diagnosis was rejected in 12 patients (32%). The 38 patients refer to the total number of patients (*n* = 43) without the two patients whose thermographic results were excluded from analysis and without the three patients with missing medical records. When the results from the FSP test for the patients whose medical records were missing were included, the overall prevalence was 71%. The results of the FSP test were positive for RP in 17 (42%) patients, negative in 19 (46%) patients, and intermediate in 5 (12%) patients. When the FSP results were compared with the clinical decisions, the diagnosis concurred in 21 patients but not in the remaining 17 patients; twelve patients with a negative FSP test result were found positive for RP based on the clinical diagnosis, whereas five patients with a positive or intermediate FSP result were found negative for RP based on the clinical diagnosis. At the 0.05 cut-off value, the thermographic diagnosis concurred with the clinical decision in 24 patients but not in the remaining 14 patients. None of the participants reported adverse events from the thermographic or the FSP procedures.

### 3.2. The Tested Thermographic Algorithm

The characteristics from the thermographic examination of the patients suspected of having RP, the subsequent application of the diagnostic algorithm, and the chosen cut-off value (0.46)—as was suggested when the algorithm was developed—resulted in a correctly confirmed diagnosis in six patients and a correctly rejected diagnosis in 12 patients, corresponding to the test performances presented in Table 3. Thus, the new thermographic algorithm was better at rejecting the diagnosis than the FSP test but was less able to identify patients with RP. Applying lower cut-off levels resulted in a similar or slightly higher accuracy for the thermographic algorithm compared with the FSP test, indicating that the thermographic method is not inferior to the FSP test in terms of diagnostic ability. The 0.05 cut-off level resulted in a slightly higher accuracy, lower sensitivity, and a considerably higher specificity for the thermographic algorithm compared with the FSP test (Table 3).

When comparing the thermographic test performance (cut-off 0.05) with the FSP test performance, the thermographic test was not shown to be superior to the FSP test for any of the performance measures. However, the sensitivity and accuracy of the thermographic test were shown to be noninferior to that of the FSP test by a 15% margin.

### 3.3. A New Algorithm

Based on the assumption that the verification of RP of a different origin requires alternative thermographic characteristics, a new algorithm was developed based on the patients diagnosed with RP by the clinical evaluation and the healthy controls.

In addition to the curve and temperature variables, age was also included in the analysis but did not contribute significantly to the prediction of RP. Gender was not included because of the small number of female patients. The binary response variable of the logistic regression model was the clinical diagnosis of RP assessed by two different senior consultants. According to the fitted logistic regression model, the combinations of predictors best able to predict the presence or absence of RP in the target population were *t_50_*, *curve type*, and *time to t_end_*. The model specifics are presented in Table 4.

The diagnostics revealed no significant deviations from the model assumptions. Five outliers (patients, as well as controls) were identified and reviewed. As no irregular values were found, they were all included in the analyses. The calibration plot and the calibration test showed that the intercept (= –0.07) and the slope (0.84) did not differ significantly from the ideal line (*p* = 0.20), indicating that the predicted probability equaled the actual (observed) probability. The AUC was 0.86 (95% confidence interval: 0.78–0.94) and the Brier score 0.17 after bootstrapping. The Brier score is an evaluation score of the logistic regression models. With values from 0 to 1, a low Brier score indicates well-calibrated predictions.

### 3.4. Cut-Off Level

Similar to the first thermographic algorithm, the new algorithm conveyed a predicted probability of having RP. The ability of the method to identify patients who have RP and the ability to correctly reject the diagnosis will, to some extent, depend on the chosen cut-off level. This is illustrated in Table 3, where the performance of the first thermographic algorithm at different cut-off levels is presented. The optimal point on the ROC curve for the new algorithm (not shown), where the sensitivity and specificity were highest, corresponded to a cut-off level of 0.28. Predicted probabilities above and below this cut-off would result in a confirmation or rejection of the diagnosis, respectively. This cut-off level yielded a sensitivity and specificity of 83% and 79%, respectively. The PPV and NPV were 67% and 90%, respectively. The accuracy was 80%. However, the new algorithm should be tested in future patients in need of verification.

## 4. Discussion

The thermographic method and resulting algorithm for verifying RP, which was previously described [8], was tested in patients suspected of having RP. When applying a cut-off level lower than the cut-off level originally chosen, the diagnostic ability of the thermographic algorithm was noninferior to that of the FSP test. In addition, the thermographic method provided more information, was less time-consuming, and seemed more agreeable to the patient than the FSP test. Furthermore, a new thermographic algorithm was proposed.

A discussion relevant to many diagnostic procedures is the choice of cut-off level. An appropriate cut-off level depends on the diagnosis in question and the clinical consequence. Is it important to identify all patients with RP, or is it clinically more useful to exclude the patients without RP? This should be discussed with the physicians who rely on this method. The sensitivity of the thermographic method when tested on the patients suspected of RP with the proposed predicted probability cut-off point of 0.05 was 65%, and the specificity was 58%. The PPV was 77%, and the NPV 44%. The same values for the FSP test were 54%, 58%, 74%, and 37%, respectively. The accuracy was 63% for the thermographic method and 55% for the FSP test. This indicates a similar diagnostic accuracy, where the strength of the thermographic method is the ability to identify patients with RP, and the physician can have confidence in a positive diagnosis.

In the literature, sensitivity and specificity for the FSP test ranged from 74% to 92% and 81% to 100%, respectively, when the patients were tested against healthy controls. The new algorithm yielded a sensitivity and specificity of 83% and 79%, respectively. Accordingly, the performance of the new algorithm—which has not yet been tested—seems to concur with the performance of the FSP test reported in the literature. The differences between studies evaluating the FSP method are likely due to differences in the study methods and population, such as the extent of cooling (temperature or local versus body cooling or both), the severity of RP in the examined patient group, and the vibration exposure level in the reference group. High sensitivities are usually found in studies applying both local and body cooling [13,14,15], low cooling temperatures [13,14,15,16], or high discriminating thresholds (60–70%). In addition, studies have argued that the sensitivity increases with the disease severity [16,17]. The patients referred to our department for examination exhibit a wide range of RP symptoms, from patients with a few and nonspecific symptoms to patients with obvious symptoms of RP. Consequently, a lower sensitivity of any method than that reported in the literature should be expected. Although the patients’ symptoms are diverse, the prevalence of RP and the symptom severity in the patients referred for examination do not seem to have changed markedly over the years of 2014–2018, based on the similar prevalence of RP in the retrospective and the prospective patient groups. As a result, there is reason to believe that the validated thermographic method and the proposed cut-off level will be useful for verifying RP in future patients referred for examination.

Compared with the previous study [8], where several parameters were significantly different on a group basis, only *time to t_end_* and *curve type* differed significantly between the confirmed patients and the controls in the present study. Most of the patients referred for examination and included in the present study were exposed to vibration and were suspected of having VWF. The considerable overlap between the patient and control groups for most of the predictors indicates that these patients are more difficult to distinguish from healthy individuals than patients with pRP. Furthermore, we found that the characteristics of pRP reported in the first algorithm differed from the characteristics of patients with RP mainly secondary to vibration exposure. This indicates that the two types of RP are two different disease entities with the same manifestations. In addition, the differences in the appropriate predictors for patients with pRP and VWF, respectively, confirm the findings of previous studies [9,10,11,18]. The *curve type*, *t_50_*, and *time to t_end_* are all dynamic parameters. If the baseline parameters are considered a surrogate for tissue damage, as was indicated in references [5,11], these findings would suggest that VWF—at least, for the included patients—is a dynamic disease without considerable tissue damage to the vessels of the fingers. Although studies have shown inflammation and remodeling of the vessels and damage to the nerve endings with resulting vascular and sensorineural dysfunction after exposure to vibration [19,20], our results are similar to the findings of von Bierbrauer et al. and House et al. [9,10].

Thermal imaging has previously been deemed useful in the management of various diseases, such as breast cancer and diabetes [21]: in breast cancer, thermal imaging seems interesting as a detection tool; in diabetes, it has proven useful in the diagnosis of long-term sequelae, such as peripheral neuropathy [22,23]. Furthermore, thermal imaging is applied in rheumatology to distinguish patients with pRP from patients with systemic sclerosis [24]. In addition to being investigated in diagnostic settings, thermal imaging has been reported as a useful screening tool for large-scale fever assessments during pandemics to prevent the dispersion of the infection [25]. The possibility of automated readings of thermographic images by machine learning methods carries great potential. However, the use of machine learning requires large image databases [26], which are not necessarily available.

Although thermographic imaging has a high potential as a medical imaging method, various conditions during image acquisition can limit its use and accuracy. The thermographic reporting of the temperature of an object is highly dependent on the object’s emissivity. Emissivity is the ability of the object to absorb and emit radiation compared to a theoretical “perfect emitter” with an emissivity of one. The emissivity of human skin is 0.98 [27], but several factors can affect its emissivity, which may result in the reporting of a lower temperature of the skin. Factors such as the presence of water [28], hair [29], or creams and lotions [28,29] on the skin can reduce its emissivity and, thereby, decrease the reported skin temperature. Theoretically, the degree of skin pigmentation might affect its emissivity, but skin color has been reported not to affect skin emissivity [27,29,30]. Accordingly, the participants in the present study wore thin plastic bags during cold provocation and were asked not to use lotions before the examination. Furthermore, we avoided placing regions of interest over the nails, as the emissivity of nails is unknown, and polished surfaces are known to have a lower emissivity than rough surfaces [31].

Along with the assessment of the test performances, other parameters such as patient discomfort, operability, and time consumption are of interest. In addition to generating more detailed information, the thermographic procedure was less technically demanding, less time-consuming, and seemed to be more pleasant for the patient than the FSP test. Indications of the patient content were based on the cooling method, time frame, and absence of episodes of RP during examination.

The patients of this study were patients referred to the department for examination. None refused participation or were excluded. Consequently, the results were highly generalizable to our department, as well as to other departments that examine unselected patients suspected of having RP mainly secondary to hand–arm vibration exposure and with similar broad symptom diversity. Furthermore, our study examined several parameters: both the baseline temperature, as well as dynamic parameters, after the cold challenge. The parameters of thermographic imaging often show considerable overlap between the patients and controls, and a combination of predictors seems necessary for diagnostic purposes.

Each participant was examined once, and test–retest reliability was not possible to assess from the current data, which poses a limitation. Although the reproducibility of dynamic thermographic imaging has been questioned, previous studies have shown good intraindividual reproducibility on both a group and an individual basis [18,32,33,34]. Nevertheless, the test–retest reliability of the present thermographic method still requires investigation. Information bias poses another limitation; the level of detail of the medical records was diverse, possibly leading to an incorrect evaluation by the readers and, thereby, an inaccurate response variable. Ultimately, this could have affected the testing of the algorithm and the fitting of the new prediction model.

## 5. Conclusions

A recently developed thermographic algorithm for verifying RP was tested in patients who were suspected of having RP and who were part of the target population. The accuracy of the thermographic algorithm was noninferior to the that of the currently applied FSP test. In addition, the thermographic method seemed easier to operate, more informative, and more agreeable to the patient than the FSP procedure. An entirely new algorithm was proposed, and the new model revealed that the predictors of pRP differed from the predictors of VWF, indicating that the two types of RP are different diseases with similar manifestations. Although the new algorithm seems useful as a diagnostic tool, its diagnostic ability must be validated in a future patient population of similar composition before it can be put into practice. Meanwhile, the original thermographic algorithm tested in the present study can replace the FSP test as a method for verifying RP. Although it is reasonable for an objective examination to assist decision-making, the results of either verification method should never be able to overrule obvious clinical symptoms of RP.

## Figures and Tables

**Figure 1 diagnostics-11-00981-f001:**
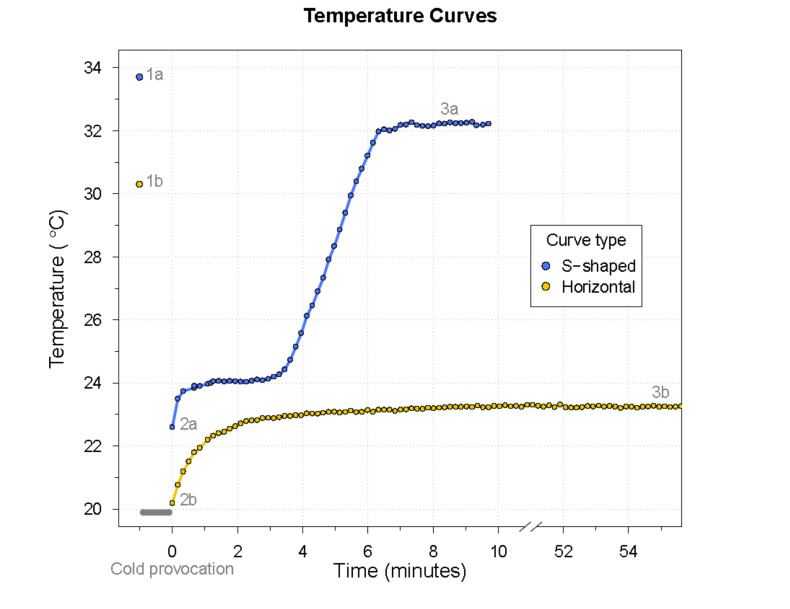
The two curve types identified during the analysis of the thermographic temperature curves. The numbers 1–3 refer to the time points of the baseline, immediately after cooling, and at the end of the thermographic examination, respectively. These time points correspond to the temperature variables baseline finger temperature (t_base_), finger temperature immediately after cooling (t_nul_), and end temperature (t_end_), respectively. The thermographic images associated with the mentioned time points are shown below the curves. The top bar shows the hands of a healthy individual (**a**), while the bottom bar presents the hands of a patient with RP (**b**). This figure is reproduced from the study, describing the recently developed algorithm [8].

**Table 1 diagnostics-11-00981-t001:** The prediction model generated from 22 patients with pRP. Reproduced from reference [8]. The time to t_end_ = time to end temperature. The t_base_ = baseline finger temperature.

Predictor	Estimate	Std. Error	Wald χ2	*p*-Value
Intercept	2.4	4.9	0.50	0.62
time to tend	0.11	0.04	2.9	0.004
tbase	−0.30	0.15	−2.0	0.04

**Table 2 diagnostics-11-00981-t002:** Clinical characteristics of the participants. Data are presented as number (%), mean (SD), or median (IQR). Raynaud’s phenomenon (RP).

		Patients Suspected of RP(*n* = 41)	Control Group(*n* = 57)	*p*-Value
Gender	Male	38 (92.7)	33 (57.9)	<0.001
	Female	3 (7.3)	24 (42.1)	
Age (Years)		56.2 (9.3)	57.8 (12.5)	0.49
Family History of RP	None	34 (82.9)	48 (84.2)	0.69
	1° Relative	5 (12.2)	5 (8.8)	
	2° Relative	1 (2.4)	0 (0.0)	
	Unknown	1 (2.4)	4 (7.0)	
Smoking	Never	12 (29.3)	35 (61.4)	<0.001
	Current	13 (31.7)	1 (1.8)	
	Former	16 (39.0)	21 (36.8)	
Smoking (Pack Years)		15 (0; 40)	0 (0; 3)	<0.001
Alcohol (Units/Week)		9 (2; 15)	4 (2; 7)	0.01
Occupation	Construction	31 (75.6)	6 (10.5)	<0.001
	Teaching and Healthcare	2 (4.9)	30 (52.6)	
	Commerce and Transportation	3 (7.3)	5 (8.8)	
	Industry	2 (4.9)	6 (10.5)	
	Finance	0 (0.0)	7 (12.3)	
	Other	3 (7.3)	3 (5.3)	

**Table 3 diagnostics-11-00981-t003:** Measures of the test performance for the finger systolic pressure (FSP) test and the evaluated thermographic algorithm with the clinical decision as the reference diagnosis (*n* = 38). The thermographic algorithm is represented by four different predicted probability cut-off levels with associated 95% confidence intervals. PPV = positive predictive value. NPV = negative predictive value.

Method		Thermographic Algorithm			FSP Test
Cut-Off	0.05	0.10	0.20	0.46	0.63
Sensitivity	65% (46–81%)	54% (35–71%)	50% (32–68%)	19% (9–38%)	54% (33–73%)
Specificity	58% (32–81%)	67% (39–86%)	75% (47–91%)	100% (76–100%)	58% (28–85%)
PPV	77% (57–90%)	78% (55–91%)	81% (57–93%)	100% (57–100%)	74% (57–86%)
NPV	44% (23–67%)	40% (22–61%)	41% (23–61%)	36% (22–53%)	37% (24–52%)
Accuracy	63% (47–77%)	58% (42–72%)	58% (42–72%)	45% (30–60%)	55% (38–71%)

**Table 4 diagnostics-11-00981-t004:** The new prediction model based on the patients diagnosed with Raynaud’s phenomenon by the clinical evaluation (*n* = 41) and the healthy controls (*n* = 57). Estimates are the respective coefficients of the model with the associated standard errors. The *p*- values show the predictors’ respective contributions to the model.

Predictor	Estimate	Std. Error	Wald χ^2^	*p*-Value
Intercept	−28.9	8.0	−3.6	<0.001
t_50_	0.92	0.27	3.4	<0.001
curve type	4.8	1.4	3.3	<0.001
time to t_end_	0.06	0.02	2.7	0.007

## Data Availability

The data presented in this study are available on request from the corresponding author.

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
