# Peer review of "Infrared Thermography as a Method of Verification in Raynaud’s Phenomenon"

_diagnostics, 2021, doi:10.3390/diagnostics11060981_

Round 1
Reviewer 1 Report
Thermographic methods have a high potential as imaging methods.
In addition, they are easy to handle and allow illustrative insights.
However, limitations and restrictions should also be considered. Which side effects can further influence the thermographic imaging results (skin moisture or scattering, pigments,...)? These issues should further be discussed.
As a diagnosis-supporting method, IR thermography should be expanded in this and similar contexts in order to demonstrate its abilities well.
Author Response
Dear Reviewer and Editor,
Thank you for your relevant comments on our manuscript. I hope that the addition of further information on the IRT method, its possibilities and limitations is sufficient. Please find my response to each of the comments listed below.
Comment 1: Thermographic methods have a high potential as imaging methods.
In addition, they are easy to handle and allow illustrative insights.
However, limitations and restrictions should also be considered. Which side effects can further influence the thermographic imaging results (skin moisture or scattering, pigments,...)? These issues should further be discussed.
Response 1: The following section has been inserted in the Discussion section, L329:
"Although thermographic imaging has high potential as a medical imaging method, various conditions during image acquisition can limit its use and accuracy. The thermographic reporting of the temperature of an object is highly dependent on the object’s emissivity. Emissivity is the ability of the object to absorb and emit radiation compared to a theoretical “perfect emitter” with an emissivity of one. The emissivity of human skin is 0.98 [27], but several factors can affect its emissivity, which may result in the reporting of a lower temperature of the skin. Factors such as the presence of water [28], hair [29], or creams and lotions [28,29] on the skin can reduce its emissivity and thereby decrease the reported skin temperature. Theoretically, the degree of skin pigmentation might affect its emissivity, but skin color has been reported not to affect skin emissivity [27,29,30]. Accordingly, the participants in the present study wore thin plastic bags during cold provocation and were asked not to use lotions before the examination. Furthermore, we avoided placing regions of interest over the nails, as the emissivity of nails is unknown, and polished surfaces are known to have a lower emissivity than rough surfaces [31]."
Comment 2: As a diagnosis-supporting method, IR thermography should be expanded in this and similar contexts in order to demonstrate its abilities well.
Response 2: The following section has been inserted in the Discussion section, L318:
"Thermal imaging has previously been deemed useful in the management of various diseases such as breast cancer and diabetes [21]: in breast cancer, thermal imaging seems interesting as a detection tool; in diabetes, it has proven useful in the diagnosis of long-term sequelae such as peripheral neuropathy [22,23]. Furthermore, thermal imaging is applied in rheumatology to distinguish patients with primary Raynaud’s phenomenon from patients with systemic sclerosis [24]. In addition to being investigated in diagnostic settings, thermal imaging has been reported a useful screening tool for large-scale fever assessment during pandemics to prevent dispersion of the infection [25]. The possibility of automated readings of thermographic images by machine learning methods carries great potential. However, the use of machine learning requires large image databases [26], which are not necessarily available."
Kind regards,
Lotte Lindberg, Corresponding author
Reviewer 2 Report
Minor corrections:
- Line 61: the word "verification" appears twice. Eventually the sentence could be modified.
- Lines 63-65: Sentence unclear. I think the authors mean "occupational exposure to vibration"?
- Line 106: FLIR is based in Oregon (US)?
Author Response
Dear Reviewer and Editor,
Thank you very much for your assessment of our manuscript. Please find enclosed my response to each of the comments.
Comment 1: Line 61: the word "verification" appears twice. Eventually the sentence could be modified.
Response 1: I agree that the sentence could be improved. I have replaced the second "verification" with "confirmation" (Line 61).
Comment 2: Lines 63-65: Sentence unclear. I think the authors mean "occupational exposure to vibration"?
Response 2: Thank you for the generous reading of this sentence. You are exactly right. I have corrected the sentence to "occupational exposure to vibration". (Line 63-65).
Comment 3: Line 106: FLIR is based in Oregon (US)?
Response 3: FLIR Systems has a corporate office in Sweden, but you are correct that their headquarters are placed in Oregon, US. Instead of "Täby, Sweden", "Wilsonville, OR, USA" has been inserted.
Kind regards,
Lotte Lindberg, Corresponding author